# Domain-Specific Text-to-Image Generation: Planning, Merging, and Replacing with Training-free LLMs

## Abstract

Diffusion-based techniques, such as Stable Diffusion, exhibit remarkable capabilities in text-to-image synthesis and editing. However, general text-to-image diffusion methods frequently fail to accurately generate domain-specific components, such as particular electrical elements in schematic circuit diagram. Lacking domain-specific knowledge, rules, and sufficient data, existing methods may struggle with resource-consumption model training. To address these limitations, we propose a novel, training-free framework for mastering domain-specific text-to-image generation, namely Planning, Merging, and Replacing (PMR). Specifically, PMR precisely generates domain-specific elements and their configurations, enabling schematic circuit diagram generation without requiring model fine-tuning. Based on the establishment of a knowledge base, PMR employs large language models (LLMs) to plan inter-component connectivity according to the requirements provided by users. PMR further utilizes LLMs to spatially arrange symbolic blocks (representing components) and their connecting wires. Subsequently, PMR has a fine-grained positional control and generates symbolic blocks and wires at designated locations. Extensive experiments demonstrate that PMR outperforms existing methods in domain-specific generation. Our work opens a potentially new avenue of automated domain-specific text-to-image generation.

## 1 Introduction

The rapid development of text-to-image diffusion models (Ho et al., 2020; Song et al., 2020; Nichol & Dhariwal, 2021; Dhariwal & Nichol, 2021) enables the generation of massive and diverse aesthetic images. However, existing open-source text-to-image diffusion models (such as Stable Diffusion (Radford et al., 2021) and SDXL (Peebles & Xie, 2023)) are primarily designed for general-purpose applications and lack specialized domain performance. Due to the absence of domain-specific training data in electrical engineering, open-source models often fail to comprehend electrical circuit terminology and concepts, resulting in irrelevant image generation. Specifically, as shown in Figure 1, in the field of schematic circuit diagram design, even if we fine-tune existing text-to-image models, they can merely generate one single electronic component and are hardly compliant with electrical regulations due to a lack of electrical knowledge. Inspired by the design of RPG (Yang et al., 2024), it is possible to utilize LLMs (Bai et al., 2023; Hurst et al., 2024; Liu et al., 2024) to rapidly acquire electrical knowledge for electric schematic circuit diagram design. Furthermore, integrating LLMs with diffusion models enables automatically generating electric schematic circuit diagrams, called schematic for brevity.

In this work, instead of training a domain-specific text-to-image model, we propose a fully automated, training-free electrical schematic circuit diagram generation framework, namely Planning, Merging, and Replacing (PMR). Our method can be extended to schematic image generation in other fields. The fully automated schematic generation framework comprises the following steps:

**Component Relationship Planning.** In this step, PMR primarily maps the components and requirements and understands the connection relationships of electrical components. By establishing an electrical knowledge base and utilizing LLMs' powerful Chain-of-Thought (CoT) planning capabilities (Zhang et al., 2023b), we plan the interconnections for components. Afterward, we use

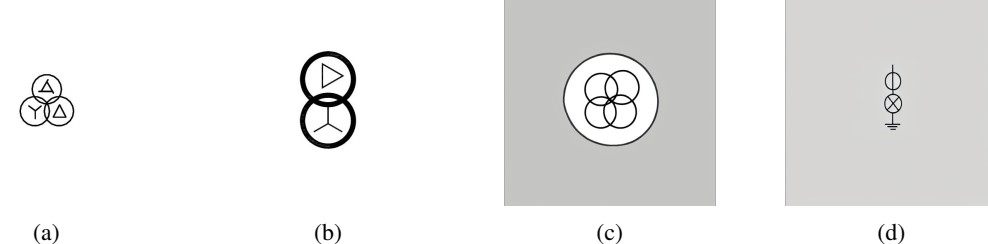

|     |     |     |     |
| :-: | :-: | :-: | :-: |
| (a) | (b) | (c) | (d) |

Figure 1: The three images (a)-(c) are **voltage transformers** generated by a fine-tuning SDXL (Peebles & Xie, 2023), producing inconsistent and unstable components. We attempt to generate an **electric transformer** and a **power-on indicator** with a parallel connection using the fine-tuned model, but it produces a series connection as shown in image (d).

black squares to represent component objects and then generate a complex initial prompt input based on the planned component connections. The component connection relationships can be manually defined, ensuring accuracy and compliance with electrical rules.

**CoT Planning for Region Division and Lines Generation.** With LLM's powerful COT reasoning capabilities, we plan the spatial positioning of each component and its connecting lines on a fixed-size drawing based on the component connection relationships obtained from the previous step. The prompt, obtained from the previous step, i.e., Component Relationship Planning, describes each individual region or object, along with its corresponding spatial locations.

**Merge Regional Diffusion.** In this step, we propose a merge regional diffusion approach to enhance the flexibility and accuracy of text-to-image generation. Specifically, we independently generate image content guided by subprompts within designated rectangular sub-regions. The regions are bound in the early denoising stage. The refinement step does not manipulate the image but instead enables interaction between regional local conditions and latent global image information across attention layers.

The novel end-to-end framework PMR enables the automatic generation of compliant schematic circuit diagrams without training, achieving full process automation. Our contributions are summarized as follows:

- We introduce a groundbreaking unsupervised schematic circuit diagram generation framework, namely Planning, Merging, and Replacing (PMR), comprising component relationship planning, merging, replacing, and generation, to maximize the synthetic capability and controllability of diffusion models.

- We leverage LLM's powerful CoT capabilities to plan component relationships while decomposing complex prompts into informative instructions for diffusion models.

- We introduce a regional diffusion approach that collaborates with LLMs to precisely generate images consistent with textual descriptions.

- Our PMR framework is user-friendly and extensible to different open-source LLMs (e.g., DeepSeek-v3 (Liu et al., 2024) and open-source text-to-image diffusion models (e.g., Flux.1-dev). Extensive qualitative and quantitative comparisons with existing open-source text-to-image methods (e.g., SDXL (Peebles & Xie, 2023)), DALL-E 3 (Betker et al., 2023)) demonstrate our superior text-guided schematic generation capabilities.

## 2 RELATED WORK

**Diffusion Models.** The foundational theory of diffusion models originated from Sohl-Dickstein et al. (2015)'s work, inspired by non-equilibrium thermodynamics. Subsequently, DDPM (Denoising Diffusion Probabilistic Models) (Ho et al., 2020) systematized this framework by employing U-Net for noise prediction and simplifying the training objective. Recently, diffusion models expanded into multimodal domains. Stable Diffusion v3 (Esser et al., 2024) replaces U-Net with Transformers to improve scalability. Recently, in text-to-image generation, quality and consistency

have been further enhanced through diverse approaches, including SDXL (Peebles & Xie, 2023), DALL-E 3 (Betker et al., 2023), and Flux.1-dev. However, when confronted with complex textual prompts, text-to-image models struggle to accurately generate images consistent with the text, particularly when the number of objects, their attributes, and spatial relationships are intricate and diverse.

**Compositional Diffusion Generation.** The evolution of controllable text-to-image models has progressed from basic generation to refined control. For example, ControlNet (Zhang et al., 2023a) achieved fine-grained spatial control over image generation by incorporating external control signals such as edge maps and depth maps. Meanwhile, DreamBooth (Ruiz et al., 2023) enabled theme-driven personalized generation through fine-tuning pre-trained models. StructureDiffusion (Feng et al., 2022) is a diffusion model focused on precisely controlling the overall layout and structure of generated images. Promptist (Hao et al., 2023) is a model designed to optimize and enhance the effectiveness of text prompts. Instancediffusion (Wang et al., 2024) enables finer-grained instance-level control. It supports specifying the position of each specific instance within an image through multiple formats (e.g., bounding boxes, masks, points, doodles) and combines free text to describe instance attributes. Concurrently, models like ReCo (Yang et al., 2023) and GLIGEN (Li et al., 2023) further explored precise manipulation of spatial layout and attributes through region-level control conditions such as bounding boxes and keypoints. Models like SDXL and Stable Diffusion v3 further integrate Transformer architectures to enhance performance and expand application scope.

**Specialized Diffusion Models.** Although control-based methods demonstrate robust performance, collecting training data is time-consuming and labor-intensive. To address these challenges, model-free training approaches have been proposed. Mulan (Li et al., 2024) is a training-free multimodal large language model agent that progressively generates multi-object images adhering to spatial relationships and property bindings. RAG (Chen et al., 2024) is an untrained region-aware text-to-image generation method. It ensures precise execution of regional prompts through hard region binding while enhancing inter-regional harmony via soft refinement. RPG (Yang et al., 2024) is a training-free text-to-image generation/editing framework. It leverages the chain-of-thought reasoning capability of multimodal large language models (MLLMs) to decompose complex prompts into sub-regional tasks, achieving compositional generation through complementary regional diffusion. These approaches are designed for general-purpose scenarios. Generating domain-specific images directly from foundational models is challenging since these models lack training on domain-specific data. In this work, we propose the zero-shot generation framework, specifically applied to schematic circuit diagram generation.

**Domain-Specific Diffusion Models.** While general text-to-image models focus on photorealism and broad adaptability, domain-specific models are tailored for structured tasks. For instance, in the realm of layout generation—which shares topological similarities with schematic design—models like LayoutDM(Inoue et al., 2023) and LayoutDiffusion(Zheng et al., 2024) employ diffusion processes to generate discrete layout elements. Unlike these supervised methods that require training on domain-specific datasets (e.g., PubLayNet(Zhong et al., 2019)), our PMR framework offers a training-free alternative, leveraging the planning capabilities of LLMs to adapt foundation models to specialized domains like electrical schematics and document layouts.

## 3 METHOD

### 3.1 OVERVIEW OF PROPOSED

In this section, we introduce our automated, training-free schematic generation framework as shown in Figure 2. Given the required components for the schematic, our framework leverages historical schematic knowledge to plan the placement of each component and the connection metrics among them. PMR subsequently generates components and connection lines using Merge Regional Diffusion, ultimately producing a schematic circuit diagram that meets requirements. In the following, we illustrate the three key steps: relationship planning, region planning, and generation.

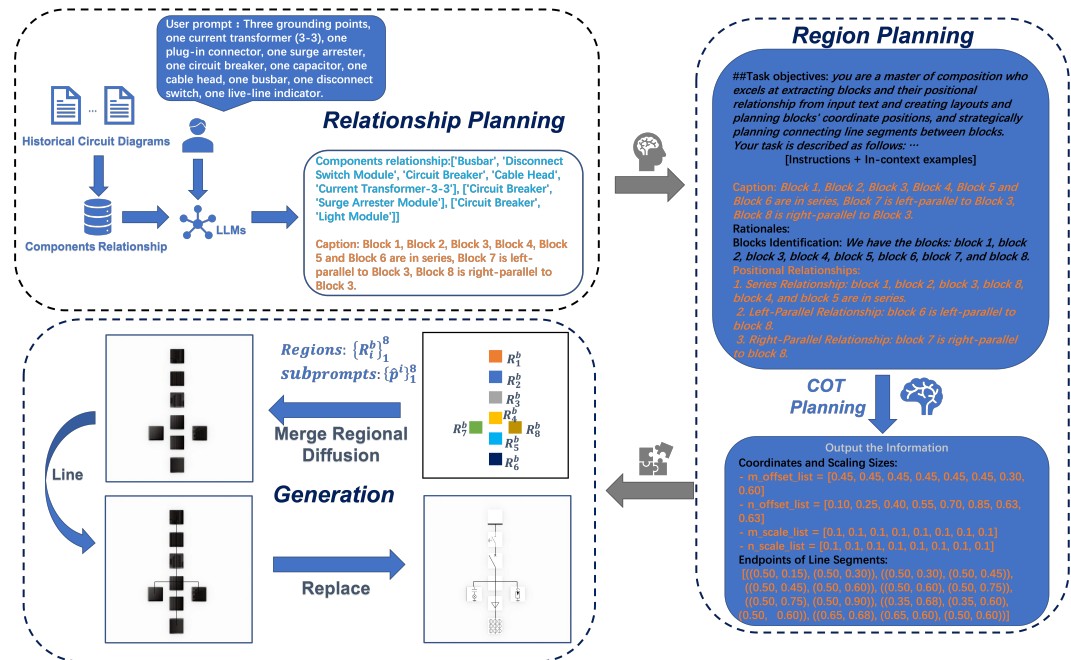

Figure 2: Overview of our PMR framework.

## 3.2 COMPONENT RELATIONSHIP PLANNING

Given the required quantity and types of components, to generate schematics, PRM should first define the component connections. As existing open-source diffusion models and LLMs lack knowledge of electrical diagram design, we establish an electrical knowledge base to support training-free schematic generation. Firstly, we recognize various components using an object detection model. The total number of component types is 30, with connection relationships summarized into three types: series, parallel, and cross nodes. The relationships are depicted in Figure 4. By employing a random walk method (Spitzer, 2001) to derive all component connection relationships from historical schematics. We systematically analyzed and summarized over 4,000 schematics. Subsequently, leveraging the LLM's powerful CoT planning capabilities alongside the knowledge base derived from historical schematics, our system plans the optimal connection relationship tree for all components. The relationship tree satisfies all the requirements and specific constraints, and defines the connection relationships of each component, i.e., whether it connects to others. Finally, we summarize the optimal connection relationship tree into a complete prompt. The LLM prompt is detailed in the appendix A.2.

We constructed a knowledge base of component connection relationships based on over 4,000 historical schematics, as shown in Figure 3. To automatically and rapidly build the knowledge base, we train an object recognition model to identify all components and their positions from historical schematics. To accelerate component relationship extraction among thousands of schematics, we modified the random walk algorithm. We utilize the tree structure to quickly determine the connection relationships between components. The improved random walk method is applied to derive the component connection relationship tree. To obtain a structured knowledge base, we define series, parallel, and cross-node connections as three types of relationship denoted as a composed relationship list:

$$\{Sr\,(z_i, z_j)\,, Pr\,(z_i, z_j, z_k)\,, Fr\,(z_i)\} \tag{1}$$

where $i, j, k \leq n$. We summarize the component connection relationship tree into these three structured relationship lists to form the final knowledge base.

In the CoT Planning component relationship step, we carefully craft task instructions and contextual examples. We leverage the powerful chain-of-thought reasoning capabilities of LLMs to achieve planning of component connection relationships. The input $\boldsymbol{p}^b$ consists of a sequence of components,

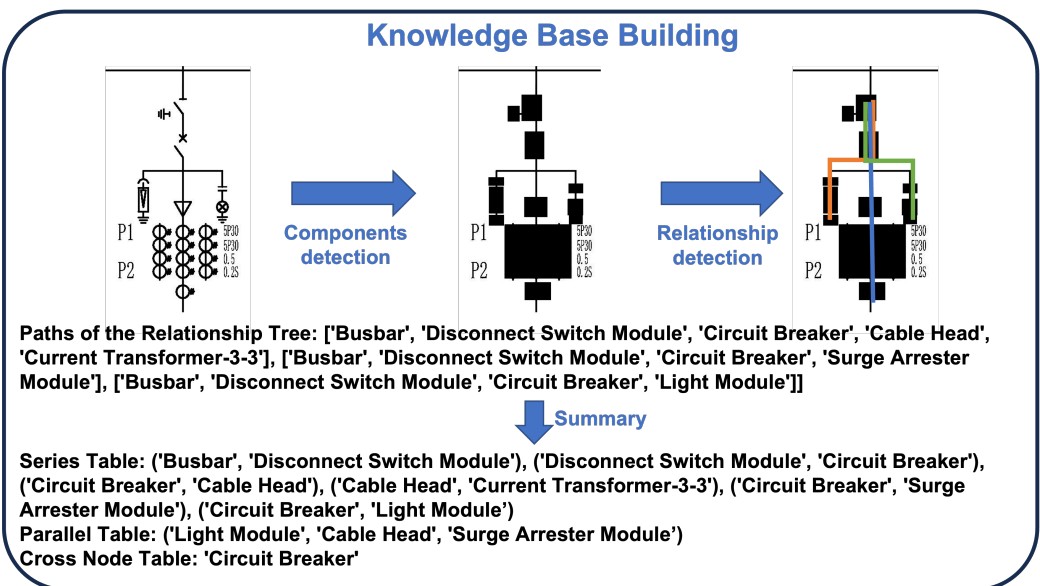

Figure 3: The entire process of building a knowledge base.

denoted as:

$$\{z_i\}_{i=1}^n = \{z_1, z_2 \cdots z_n\} \subseteq P^b \tag{2}$$

where $n$ denotes the number of components, $z_i$ denotes one of the all components. Through carefully designed task instructions for component connection planning and detailed contextual examples that contain explicit reasoning steps, we guide the large model to perform accurate inference following the examples. We then use GPT-4o(Hurst et al. (2024)) or other LLMs such as DeepSeek-v3, Qwen(Bai et al. (2023)), to output all connection relationship of the schematic circuit diagram. Subsequently, we consolidate all obtained component connection relationship into a comprehensive prompt $\hat{P}^b$. First, we abstract all components as uniformly sized black squares denoted as:

$$\{\hat{z}_1, \hat{z}_2 \cdots \hat{z}_n\} = Rename\left(\{z_i\}_{i=1}^n\right) \tag{3}$$

where $n$ denotes the number of components, $\hat{z}_i$ denotes Block $i$. Simplifying the positioning process, as existing text-to-image foundational models cannot generate diagrams for specialized electrical components, we initially generate squares, which are later replaced with corresponding components. Consequently, component names in the prompt will be replaced with labels like Block 1, Block 2, etc, denoted as:

$$\{\hat{z}_i\}_{i=1}^n = \{\hat{z}_1, \hat{z}_2 \cdots \hat{z}_n\} \subseteq \hat{P}^b \tag{4}$$

Simultaneously, we simplify and consolidate all component connection relationships to facilitate subsequent positioning planning and comprehension. All series-connected components are grouped into a single "series" while parallel-connected components are categorized as left-parallel or right-parallel. The final prompt is denoted as:

$$P^c = Recaption\left(\hat{P}^b\right) \tag{5}$$

### 3.3 CoT Planning for Region Division and Lines

To plan the position of each component and the layout of connecting lines between components, we again leverage the powerful CoT reasoning capabilities of LLM. The LLM prompt is detailed in the appendix A.3. Existing diffusion models frequently omit details, failing to accurately match the information described in the text. To address this issue, we decompose the original complex prompt containing multiple objects into basic descriptor subsets for each individual region or object, along with their corresponding spatial positions. This process can be accomplished using LLMs or through manual definition. We continue to utilize GPT-4o, though other large language models can be substituted.

Since the previous step yields a complete connection-relationship prompt $P^c$ encompassing all black square objects and their interconnections, this step decomposes that comprehensive prompt into basic descriptor subsets for each square object and their corresponding spatial positions. For each block object's basic descriptor subprompt $p_i$, since we've abstracted components as black squares, the rectangular region corresponding to each block object in the text-to-image diffusion model prompt is uniformly defined as "black square block" occupying the entire rectangular area, which can be denoted as:

$$\{p_1, p_2 \cdots p_n\} = Recaption\left(\{\hat{z}_i\}_{i=1}^n\right), \forall i \in [1, n] \quad p_i = {''black\ square\ block''} \tag{6}$$

For each block's spatial position, we define a spatial position as rectangular region. Each rectangular region's information is described by four parameters to illustrate its position and size, where $m_{offset}^i$ and $n_{offset}^i$ denote the $x$ and $y$ coordinates of its top-left corner vertex, $m_{scale}^i$ and $n_{scale}^i$ denote the width and height of the region. Concretely, we assign each subprompt $p_i$ to specific region $R_i^b = \left\{m_{offset}^i, n_{offset}^i, m_{scale}^i, n_{scale}^i\right\}$, and each rectangular region is mutually non-overlapping, as shown in Figure 5. Thus, we have

$$\left\{R_i^b\right\}_{i=1}^n = \left\{R_1^b, R_2^b \cdots R_n^b\right\} \subseteq H \times W \tag{7}$$

where $H$ denotes the height of the schematic image and $W$ denotes the width.

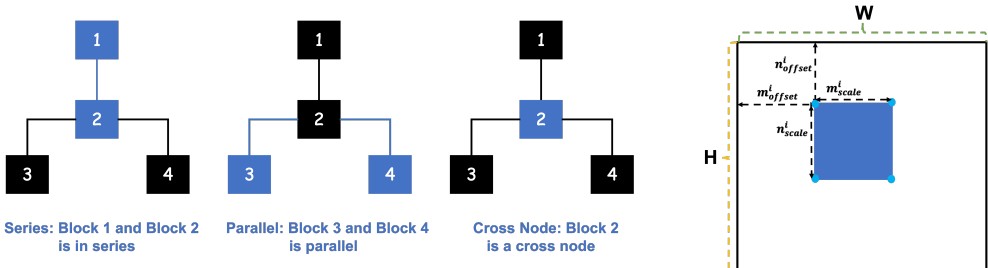

Figure 4: Three types: Series, Parallel, and Cross.    Figure 5: Parm. in Region Division.

We leverage the LLM's powerful CoT reasoning capabilities, combined with detailed contextual examples, to plan the number and positions of complementary rectangular regions based on the component connection relationships prompt $P^c$. In addition, we can precisely control the size of black squares and the spacing between them by adjusting the dimensions of rectangular regions and the gaps between them, as shown in Figure 10.

Afterward, we plan the generation regions for the lines on the schematic circuit diagram. There are two types of connections between components: straight segments (primarily for series connections) and angled segments (primarily for parallel connections). For straight segments, two endpoint coordinates are used. For angled segments, two endpoints and one bend point are denoted as:

$$straight\ segments : ((a_1, b_1), (a_2, b_2)), \quad angled\ segments : ((a_1, b_1), (a_2, b_2), (a_3, b_3)). \tag{8}$$

We combine each component block's position on the diagram with the complete prompt. We use straight, left-angled, and right-angled segments for series connections, left parallel connections, and right parallel connections, respectively. The detailed information is in the appendix A.4. The connection planning is also based on the LLM. The LLM prompt is detailed in the appendix A.3.

### 3.4 MERGE REGIONAL DIFFUSION

Recent work on complementary region diffusion has adjusted cross-attention masks or layouts to facilitate compositional generation. However, these methods primarily rely on simply stacking latent factors, leading to conflicts in overlapping regions and ambiguous results. To this end, we introduce a novel approach called Merge Region Diffusion for regional generation and image synthesis, as shown in Figure 6. We extract non-overlapping complementary rectangular regions and apply a merging step to achieve high-quality synthetic generation. We employ the new merge region diffusion method to generate blocks at specified rectangular positions, $\left\{R_i^b\right\}_{i=1}^n$ within the schematic,

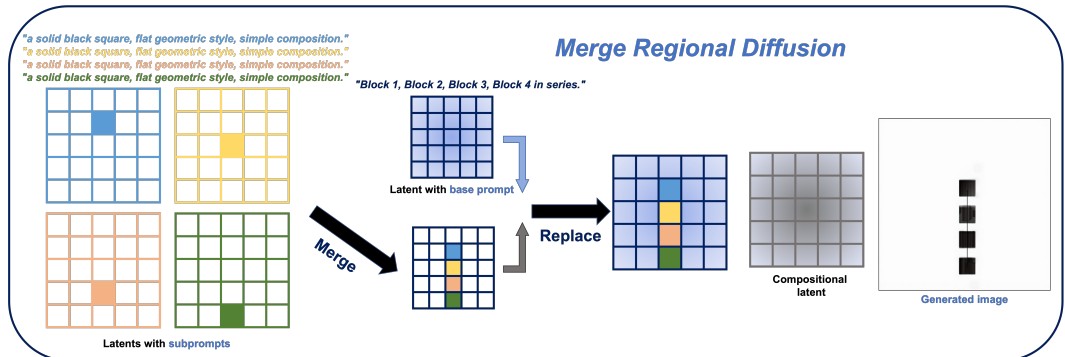

Figure 6: Overview of Merge Regional Diffusion.

ensuring background continuity. In our case, the background is pure white, making background continuity less critical. This approach decouples the schematic generation process into constructing individual regions and refining details. Involving the decomposition of original, complex prompt $P^c$ containing multiple component objects into basic subsets of descriptors $\{p_i\}_{i=1}^n$ for each distinct region or object, along with their corresponding spatial positions $\{R_i^b\}_{i=1}^n$. Each region is then processed individually with its fundamental descriptors, bound only during the early stages of denoising to ensure accurate property representation and entity localization. In the meantime, to ensure correct responses to regional prompts and reduce object omissions when the number of regions or objects increases, we apply regional hard binding in the early stages of the denoising process. The formulation is as follows:

$$z_{t-r} = z_{t-r+1} - \epsilon_\theta\left(z_{t-r+1}, y\right) = softmax\left(\frac{\left(W_Q \cdot \phi\left(z_{t-r+1}\right)\right)\left(W_K \cdot \psi\left(P^c\right)\right)^T}{\sqrt{d}}\right) W_V \cdot \psi\left(P^c\right)$$
(9)

$$\hat{z}_{t-r}^i = \hat{z}_{t-r+1}^i - \epsilon_\theta\left(\hat{z}_{t-r+1}^i, \hat{y}^i\right) = softmax\left(\frac{\left(W_Q \cdot \phi\left(\hat{z}_{t-r+1}^i\right)\right)\left(W_K \cdot \psi\left(\hat{p}^i\right)\right)^T}{\sqrt{d}}\right) W_V \cdot \psi\left(\hat{p}^i\right)$$
(10)

where $i \in [1, n]$, $n$ is the number of regions, $r$ is one of the early steps within the denoising process. $\epsilon_\theta$ is the noise predicted. And image latent $z_t$ is the query, prompt $P^c$, and each subprompt $\hat{p}^i$ works as a key and value. $W_Q, W_K, W_V$ are linear projections and $d$ is the latent projection dimension of the keys and queries. Then, we shall proceed with replacing the base latent $z_{t-r}$ with the generated latent $\left\{\hat{z}_{t-r}^i\right\}_{i=1}^n$ and merging, according to their assigned region numbers from 0 to $n$. Specifically, we perform text encoding on $P^c$ and $\hat{p}^i$ to obtain $y$ and $\hat{y}^i$. Individual latent $\hat{z}^i$ is text-conditioned on $\hat{y}^i$, while the origin latent $z$ is conditioned on the complex prompt $P^c$. For each denoising step, we bind $\hat{z}_{t-r}^i$ to the latent space in the rectangular area given by $R_i^b$ as follows:

$$z_{t-r} = Merge\ and\ Replace\left(z_{t-r}, \hat{z}_{t-r}^i, R_i^b\right)$$
(11)

where $Merge\ and\ Replace\ (\cdot)$ denotes the process of pasting individual latents back into the corresponding regions of the original latent. Binding is performed only during the early stages of the denoising process. We find that binding over several steps suffices to achieve regional integrity, whereas binding over all steps leads to sharp visual boundaries between adjacent regions or poor interactivity.

## 4 EXPERIMENTS

### 4.1 EXPERIMENT SETTING

**Implementation Details.** Our PMR framework is general and extensible, we can incorporate arbitrary LLM architectures and diffusion backbones into the framework. In our experiment, we choose GPT-4o as the recaptioner and CoT planner, and use Flux.1-dev as the base diffusion backbone

to build our PMR framework. Concretely, in order to trigger the CoT planning ability of LLMs, we carefully design task-aware templates and high-quality in-context examples to conduct few-shot prompting. As the first framework in the electrical engineering domain to utilize diffusion models for automated diagram generation, our novel end-to-end electrical diagram generation framework faces a critical limitation: existing open-source text-to-image models cannot recognize the specialized terminology of electrical components, rendering them incapable of generating electrical components or diagrams. To demonstrate the power of our approach, we conducted experimental comparisons between the core Merge Regional Diffusion module within our full-process electrical diagram generation framework and various existing open-source text-to-image models. All experiments are conducted on a single A800 GPU.

**Compared Methods.** To comprehensively evaluate the generation quality, we compare our PMR with several state-of-the-art text-to-image approaches, including: Stable Diffusion v1.5, Stable Diffusion v3.5, SDXL, DALL-E 3, Pixart-$\alpha$-ft(Chen et al. (2023)), RPG, and Flux.1-dev.

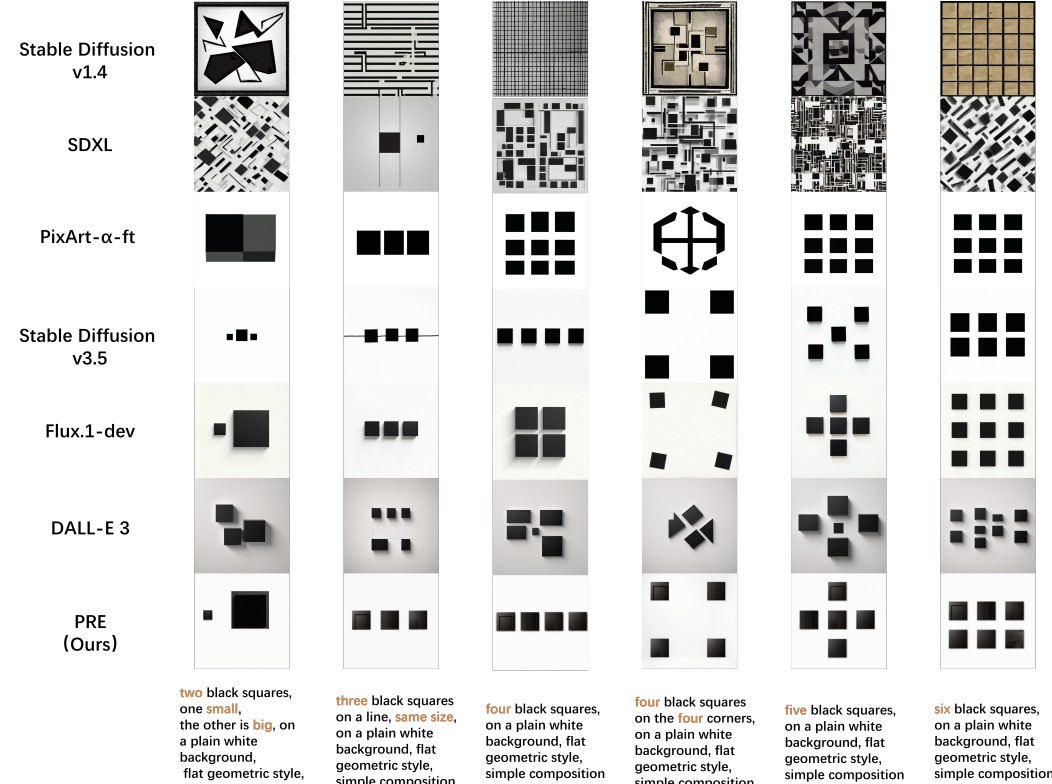

Figure 7: Qualitative comparison between our PMR and SOTA text-to-image models.

## 4.2 Main Results

**Quantitative Comparison.** We compare with previous SOTA text-to-image models in three main compositional scenarios: (i) Attribute Binding. Each text prompt in this scenario has multiple attributes that bind to different entities. (ii) Numeric Accuracy. Each text prompt in this scenario has multiple entities sharing the same class name. The number of each entity should be greater than or equal to two. (iii) Complex Relationship. In this scenario, each test prompt contains multiple component block objects with different attributes (e.g., block size) and relationships (e.g., spatial and location). We primarily test whether each model can correctly generate the quantity of component block objects, their individual size attributes, the spatial relationships between component blocks, and even the precise gap sizes between them. Finally, we also test whether the background of the generated drawings is continuous and uniform, as shown in Figure 7. Table 1 presents that PMR

Table 1: Comparison of alignment evaluation on T2ICompBench. The best results are highlighted in **bold**.

| Method | Attribute Binding | | | Object Relationship | | Complex↑ |
|---|---|---|---|---|---|---|
| | Color↑ | Shape↑ | Texture↑ | Spatial↑ | Non-Spatial↑ | |
| Stable Diffusion v1.4 | 0.3765 | 0.3576 | 0.4156 | 0.1246 | 0.3079 | 0.3080 |
| SDXL | 0.5879 | 0.4687 | 0.5299 | 0.2133 | 0.3119 | 0.3237 |
| Pixart-$\alpha$-ft | 0.6690 | 0.4927 | 0.6477 | 0.2064 | 0.3197 | 0.3433 |
| DALL-E 3 | 0.7785 | **0.6205** | 0.7036 | 0.2865 | 0.3003 | 0.3773 |
| RPG | 0.7476 | 0.5640 | 0.6724 | 0.4017 | 0.3032 | 0.3702 |
| Flux.1-dev | 0.7680 | 0.5078 | 0.6195 | 0.2606 | 0.3078 | 0.3650 |
| **PRE(Ours)** | **0.7849** | 0.5926 | **0.7064** | **0.4687** | **0.3206** | **0.4056** |

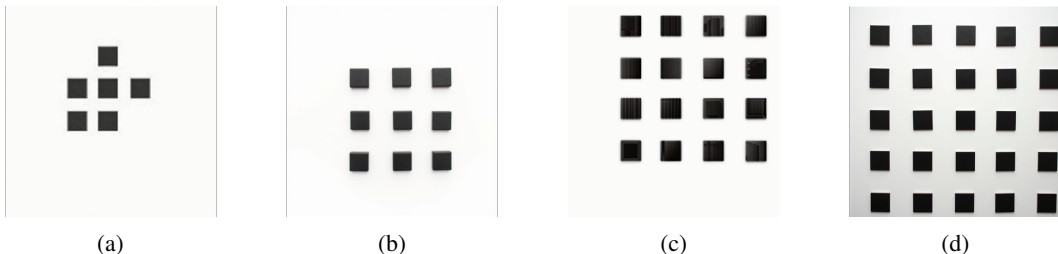

|  (a)  |  (b)  |  (c)  |  (d)  |

Figure 8: **Analysis of Numeric Accuracy.** From images (a)-(d), our PMR generates with 6, 9, 16 and 25 component blocks.

outperforms competitors in key aspects such as attribute binding, object relationships, and complex composition.

**Analysis of Numeric Accuracy.** Regarding numeric accuracy, our comparative testing reveals that for the latest open-source text-to-image models, when the text prompt contains more than six component blocks, these models generally fail to accurately generate the specified number of blocks—often producing either more or fewer. Our Merge Regional Diffusion generation module can control the creation of up to 25 component blocks anywhere on the drawing, provided they do not overlap. The Figure 8 displays drawings generated with 6, 9, 16, and 25 component blocks, each with a side length of 0.1 and manually specified positions. This demonstrates the superiority of our method, enabling precise control over the diffusion generation of each object to ensure no element is omitted.

**Analysis of Size Accuracy.** Regarding size accuracy, the latest open-source text-to-image models can only generate component blocks with vague relative sizes like "big" or "small", not even able to generate these as shown in Figure 7. Our Merge Regional Diffusion generation module enables precise control over component block dimensions, generating sizes ranging from 0.1 to 1.0. This level of granular control is crucial for electrical diagram generation. As shown in Figure 10 in appendix A.5.

**Analysis of Location Accuracy.** Regarding location accuracy, the latest open-source text-to-image models can only control relative positions such as left, right, top, or bottom. They cannot precisely control absolute positions on a drawing. Our Merge Regional Diffusion module, however, enables fine-grained control over the placement of component blocks anywhere on the entire drawing. This level of precision is crucial for electrical diagram generation. As shown in Figure 8 and Figure 10 in appendix A.5.

## 5 CONCLUSION

In this paper, we introduce PMR (Planning, Merging, and Replacing) , a novel training-free approach for automated electrical schematic generation. We address the limitations of general-purpose text-to-image models like Stable Diffusion and SDXL in understanding technical electrical terminology by strategically combining LLMs' CoT planning capabilities with a specialized regional diffusion process. PRM ensures precise alignment with electrical engineering standards and outperforms existing methods in attribute binding, numeric accuracy, and complex relationship generation.

**Limitation and Future Work.** Our current PMR method is limited by its relatively slow reasoning generation speed, which represents a key area for future improvement to accelerate inference. Additionally, while the framework has been applied in the electrical domain, it could be extended to other specialized fields for schematic generation in the future.

## 6 ETHICS STATEMENT

We confirm that this work complies with the ICLR Code of Ethics. The datasets used in our experiments are either publicly available under appropriate licenses or released with explicit consent. No personally identifiable or sensitive information was collected or disclosed. We have carefully considered potential risks, including fairness, bias, and possible misuse of our methods, and we discuss limitations in the main text. Our contributions are intended for scientific and educational purposes only, and do not promote harmful applications.

## 7 REPRODUCIBILITY STATEMENT

We have made significant efforts to ensure the reproducibility of our results, in line with the ICLR reproducibility guidelines. Specifically:

- We provide detailed descriptions of datasets, preprocessing steps, model architectures, hyperparameters, training procedures, and evaluation metrics in the main text and appendix.
- All random seeds were fixed, and we report results averaged over multiple runs to account for variability.
- The source code, configuration files, and pretrained models will be released upon publication at https://anonymous.4open.science/r/PMR-1D83/README.md.
- For datasets that cannot be shared directly due to licensing or privacy restrictions, we provide acquisition instructions or synthetic substitutes.

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

## A  APPENDIX

This supplementary material is structured into several sections that provide additional details and analysis related to PMR. Specifically, it will cover the following topics:

- In Appendix A.1, we clarify the use of large language models and describe their precise role.
- In Appendix A.2, we provide the detailed prompt template of the Component Relationship Planning by the LLM.
- In Appendix A.3, we provide the detailed prompt template of the CoT Planning for Region Division and Lines Generation by the LLM.
- In Appendix A.4, we provide the detailed information on planning the generation regions for the lines on the circuit.
- In Appendix A.5, we provide the detailed information on size accuracy analysis in blocks' generation.
- In Appendix A.6, we provide qualitative examples of final schematic generation.
- In Appendix A.7, we provide generalization and cross-Domain analysis.

### A.1  USE OF LARGE LANGUAGE MODELS (LLMS)

Large language models (LLMs) were used only for minor writing assistance (e.g., grammar checking and improving readability). All research ideas, experimental design, implementation, and analysis are original contributions of the authors.

### A.2  DETAILED PROMPT TEMPLATE OF THE COMPONENT RELATIONSHIP PLANNING BY THE LLM

As stated in Section 3.2, PMR first conducts the Component Relationship Planning to leverage the LLM's powerful CoT planning capabilities alongside the knowledge base derived from historical drawings, the system plans the optimal connection relationship tree for all components. To this end, given the input prompt $p$, we prompt the LLM using the following template:

> As an expert proficient in system architecture design, you must generate the most probable connection relationship diagram (tree structure) for all components based on the historical knowledge base. The task is described as follows:

Input all components; output the most probable component connection relationship diagram.

A. Three Types of Tables in the Historical Knowledge Base:
Series Table: The first entry in the table indicates the most frequent series connection, e.g.,('Busbar', 'Circuit Breaker Isolation Switch Group'), signifying that the circuit breaker isolation switch group is a child node of the busbar.
Parallel Table: The first entry indicates the most frequent parallel relationship. For example,('Current Transformer-3-5', 'Disconnect Switch', 'Grounding', 'Busbar') shows these four components are subnodes of a specific component.
Cross Node Table: Components within have multiple subnodes, with the number of subnodes being greater than or equal to 1 and less than or equal to 3.

B. Historical Knowledge Base.

C. Strict Requirements:
1. Use only user-provided components and quantities, do not exceed specified ranges.
2. General busbars serve as root nodes.
3. Grounding points can only be leaf nodes without subnodes.
4. Strictly utilize the above historical knowledge base (prioritizing the first entry).

D. Planning Steps:
1. First determine if the node is a branch node. If yes, it has $\geq 1$ and $\leq 3$ child nodes; otherwise, it has only one child node.
2. If the node is not a branch node, select child nodes from the series table left-to-right. If the node is a branch node, select child nodes from both the series and parallel tables. Ensure all selected subnodes remain within the user-input components.
3. Repeat the above two steps until all user-input components are selected.
4. If any isolated nodes remain after final selection, connect them to the main tree using known parallel connections to form a single tree structure.

E. Based on user-input components, construct the most probable component connection tree (the primary path typically follows a tree structure containing current transformers or voltage transformers).

G. Check if the component connection tree contains specific modules. Currently, there are three types: Disconnect switch module: ('Disconnect Switch', 'Grounding'), merged these two components into a disconnect switch module. The surge arrester module is formed by connecting three components: ('plug-in connector', 'surge arrester', 'grounding') or two components: ('surge arrester', 'grounding'), merged into a single surge arrester module; the lamp module is formed by connecting three components: ('capacitor', 'lamp', 'grounding'), merged into a single lamp module.

F. The final connection relationships are output as an array in a strictly defined format.

In this way, the LLM will decompose the input prompt $p$ following the pre-defined order.

## A.3 DETAILED PROMPT TEMPLATE OF THE COT PLANNING FOR REGION DIVISION AND LINES GENERATION BY THE LLM

As stated in Section 3.3, after determining the required connection relationships for all components in the component relationship planning phase, this step involves planning the position of each component on the drawing and the layout of connecting lines between components. We again leverage the powerful CoT reasoning capabilities of LLM. To this end, given the input prompt $p$, we prompt the LLM using the following template:

you are a master of composition who excels at extracting blocks and their positional relationship from input text and creating layouts and planning blocks' coordinate positions, and strategically planning connecting line segments between blocks. Your task is described as follows:

Extract the blocks and their positional relationship from the input text, and determine how many regions should be splited and how to plan connecting line segments between blocks. For each key block identified in the previous step, use precise spatial imagination to assign each object to a specific area within the image. Each block is assigned to a region. For each block, place it in the designated square position, reasonably plan its top-left corner coordinates and scaling size relative to the entire image in accordance with positional relationship, ensuring that it does not exceed its allocated region. Additionally, any two squares must not overlap and should have gaps between them.

This layout should segment the image and how to plan connecting line segments between blocks strictly follow the method below:

a. Extract all blocks and their positional relationship, from the input text, excluding any redundancy information;

b. Determine all blocks' top-left corner coordinates(HB_m_offset, HB_n_offset) and ceter coordinates(C_m, C_n) by their positional relationship and the endpoints of line segments between blocks by blocks' ceter position;

c. Determine all blocks' scaling sizes. From step a;

d. Output all blocks' top-left corner coordinates and scaling sizes;

e. Output all lines' endpoints coordinates.

In this way, the LLM will decompose the input prompt $p$ following the pre-defined order.

## A.4 DETAILED INFORMATION OF PLANNING THE GENERATION REGIONS FOR THE LINES ON THE CIRCUIT

As stated in Section 3.3, detailed information of planning the generation regions for the lines on the circuit has three types, for series connections, use straight segments; for left parallel connections, use left-angled segments; for right parallel connections, use right-angled segments, as shown in Figure 9.

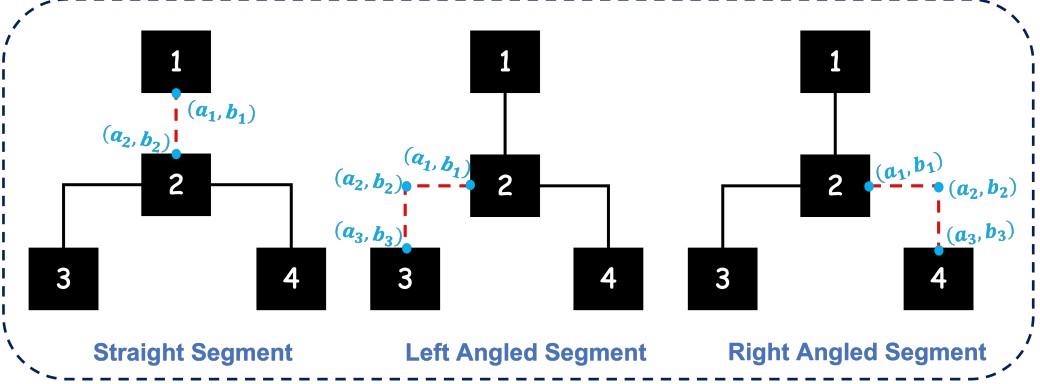

Figure 9: Types of connections between components.

## A.5 DETAILED INFORMATION OF ANALYSIS OF SIZE ACCURACY IN BLOCKS' GENERATION

Our Merge Regional Diffusion generation module enables precise control over component block dimensions, generating sizes ranging from 0.1 to 1.0. Figure 10 shows our method can generate all kinds of size of block at any location on the drawing.

## A.6 QUALITATIVE EXAMPLES OF FINAL SCHEMATIC GENERATION

In this section, we provide additional qualitative results to supplement the quantitative analysis. Figure and Figure display two generated samples of classic electrical circuit diagrams. These images illustrate the complete PMR pipeline, confirming that our method effectively translates text prompts into topologically correct and visually standard schematics by replacing the intermediate diffusion-generated placeholders with accurate domain-specific component symbols.

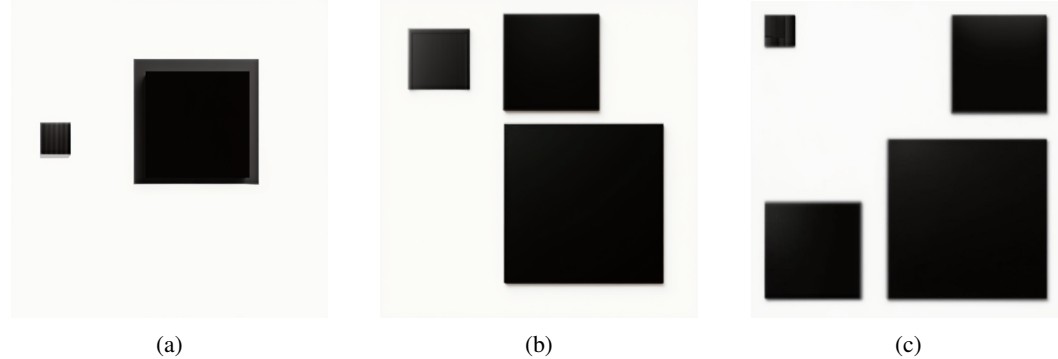

(a)          (b)          (c)

Figure 10: **Analysis of Size Accuracy.** From images (a)-(c), our PMR generates three types of different sizes of component blocks.

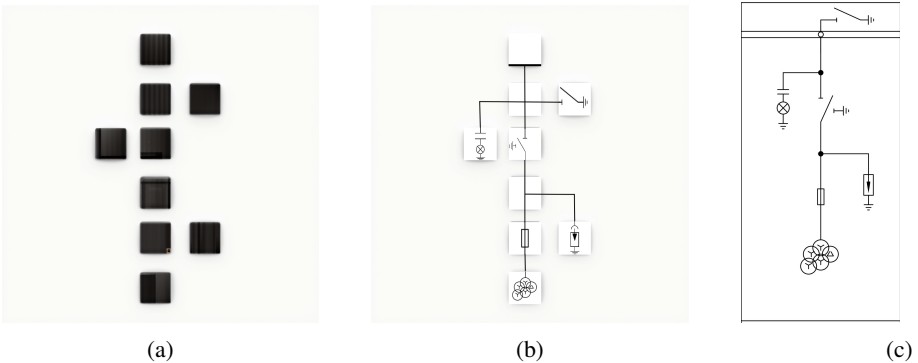

(a)          (b)          (c)

Figure 11: Image (a) displays the intermediate spatial layout where components are represented as black block placeholders. Image (b) shows the final synthesized circuit diagram after substituting the blocks with domain-specific electrical symbols, demonstrating high structural alignment with the original ground truth schematic shown in (c).

## A.7 GENERALIZATION AND CROSS-DOMAIN ANALYSIS

To demonstrate that PMR is a versatile, domain-agnostic framework rather than a specialized tool for specific schematics, we extended our evaluation to two additional scenarios:

**Controllable Layout Generation:** We applied the PMR pipeline to document layout generation using the PubLayNet dataset. Figure visualizes the results. In contrast to supervised methods like LayoutDM(Inoue et al., 2023), which rely on training generative models on labeled layout data, PMR achieves high-quality layout synthesis in a strictly training-free manner. Furthermore, by utilizing LLMs for the Planning phase, our method introduces a higher degree of layout diversity and logical coherence, avoiding the mode collapse issues often observed in trained models.

**Generalization to Diverse Schematic Styles:** We further validated the robustness of our framework by applying it to a broader, more universally styled circuit diagram dataset, distinct from the private industrial dataset used in our main experiments. As shown in Figure, PMR successfully generalizes to these new visual standards without requiring model fine-tuning or LoRA adaptation. This confirms that the "Planning, Merging, and Replacing" paradigm effectively disentangles structural logic from visual style, making it adaptable to various graphical domains.

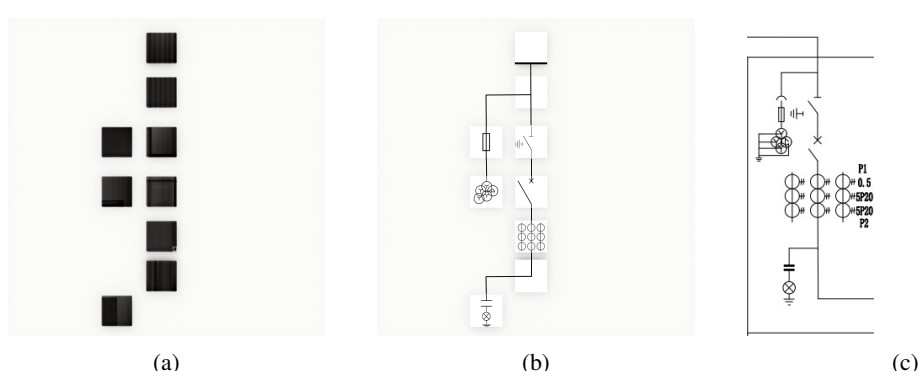

(a)                                          (b)                                          (c)

Figure 12: Image (a) displays the intermediate spatial layout where components are represented as black block placeholders. Image (b) shows the final synthesized circuit diagram after substituting the blocks with domain-specific electrical symbols, demonstrating high structural alignment with the original ground truth schematic shown in (c).

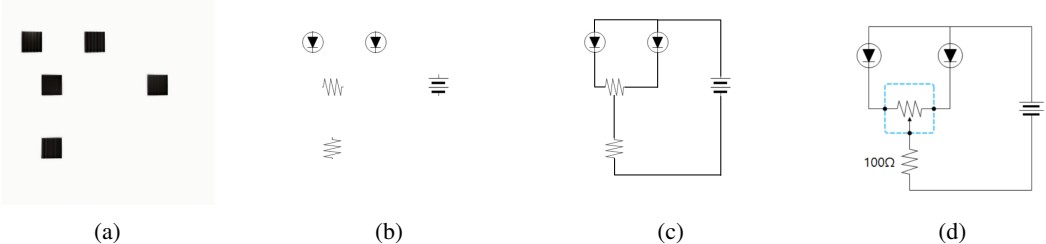

(a)                          (b)                          (c)                          (d)

Figure 13: **Visualization of the stepwise generation process.**(a) The intermediate spatial layout generated with black square placeholders. (b) The schematic after the Replacing phase, where placeholders are substituted with actual electrical component symbols. (c) The final generated schematic circuit diagram. (d) The original historical schematic (Ground Truth) for reference. **User Input:**"One Battery, Two LED, Two Resistor."

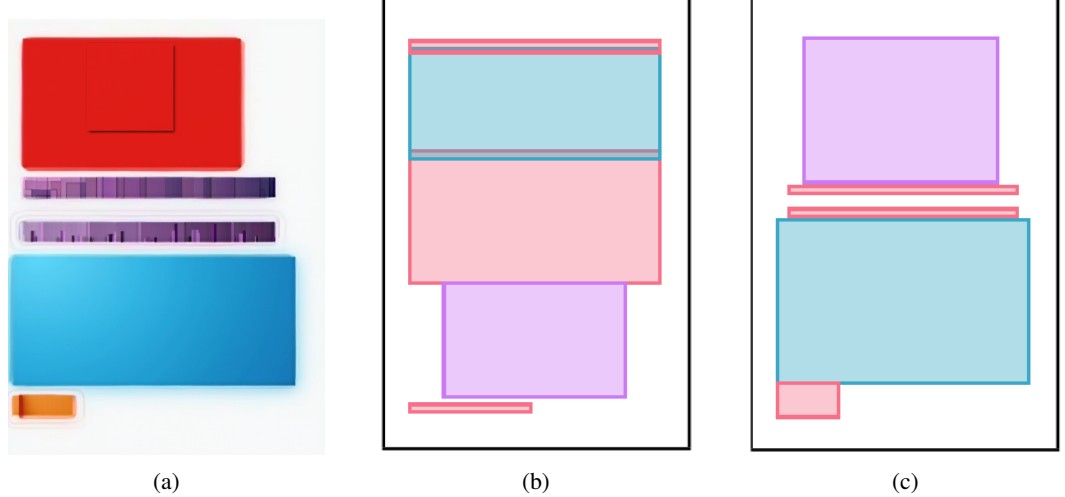

(a)                                          (b)                                          (c)

Figure 14: **Qualitative comparison of layout generation on the PubLayNet dataset.** (a) The document layout generated by our proposed PMR framework (training-free). (b) The layout generated by the baseline method, LayoutDM (requires training). (c) The original ground truth layout for reference. **User Input:**"Three Text, one Table, one Figure."

