# OpenReview forum: "Domain-Specific Text-to-Image Generation: Planning, Merging, and Replacing with Training-free LLMs"
_ICLR.cc/2026/Conference — Submitted to ICLR 2026_

### Official Review · Reviewer_xMiG · 2025-10-18

**Soundness:** 2
**Presentation:** 2
**Contribution:** 2
**Rating:** 4
**Confidence:** 3

**Summary:**

This paper tackles the failure of text-to-image models in generating circuit diagrams. It proposes PMR, a novel training-free framework. PMR uses LLMs to plan component connectivity and layout, then guides a diffusion model for fine-grained, spatially-controlled generation, avoiding the need for resource-intensive fine-tuning.

**Strengths:**

Targeting electrical circuit generation, the authors' proposed method of Planning, Merging, and Replacing advances beyond several existing T2I baselines.

**Weaknesses:**

1.  The title of the paper claims the contribution is "Domain-Specific", yet the methodology and experiments focus exclusively on a single domain (e.g., electrical circuit generation). This is confusing and potentially misleading. If the method's generality has not been substantiated, the title and claims should be narrowed to specifically reflect its application to electrical circuit generation.

2.  Section 3.4 is confusing to me; can you explain the complete denoising process? Please also clarify the role of the softmax operation and the meaning of the function \(\phi\) in Equations 9 and 10. Furthermore, have you compared your method with layout-to-image diffusion models (e.g., LayoutDiffusion [1], LayoutDM [2])? Such a comparison would help strengthen the persuasiveness of your work.

3.  It seems the experimental dataset used for evaluation is not clearly specified. Please clarify whether a publicly available benchmark or a private dataset was used.

**References**

[1] Zheng, G., et al.: LayoutDiffusion: Controllable diffusion model for layout-to-image generation. In: Proceedings of the IEEE/CVF Conference on Computer Vision and Pattern Recognition (2023)

[2] Inoue, N., et al.: LayoutDM: Discrete diffusion model for controllable layout generation. In: Proceedings of the IEEE/CVF Conference on Computer Vision and Pattern Recognition (2023)

**Questions:**

Please see Weaknesses section

---

> ### Author Response · Authors · 2025-11-26
> **General Response: Clarification on Domain Generality, Methodological Details, and Experimental Data**
>
> **Response to Reviewer**
>
> We sincerely thank the reviewer for their constructive feedback. We have addressed the specific concerns regarding the paper's scope, technical implementation details, and experimental datasets below.
>
> **Weakness 1: The title of the paper claims the contribution is "Domain-Specific", yet the methodology and experiments focus exclusively on a single domain (e.g., electrical circuit generation). This is confusing and potentially misleading. If the method's generality has not been substantiated, the title and claims should be narrowed to specifically reflect its application to electrical circuit generation.**
>
> **A1:** We agree that demonstrating applicability beyond a single domain is crucial. We have expanded our evaluation in the new **Appendix A.6** to include two additional scenarios verifying PMR's generality:
> 1.  **Controllable Layout Generation:** Applied to the **PubLayNet** dataset. Unlike supervised methods (e.g., LayoutDM), PMR is **training-free** and leverages LLM planning to generate diverse layouts without fine-tuning.
> 2.  **Universal Circuit Styles:** Tests on a public dataset of universally styled circuits confirm PMR generalizes to varying visual styles without domain-specific training.
>
> **Weakness 2.1: Section 3.4 is confusing to me; can you explain the complete denoising process? Please also clarify the role of the softmax operation and the meaning of the function ($\phi$) in Equations 9 and 10.**
>
> **A2.1:**
> **Complete Denoising Process:**
> 1.  **Decomposition:** Global prompt decomposed into regional subprompts.
> 2.  **Hard Binding (Early Stage):** For the initial $r$ steps, latents are generated independently for each region conditioned on subprompts.
> 3.  **Merge and Replace:** Regional latents are pasted back into the global latent at specific coordinates.
> 4.  **Refinement (Late Stage):** Explicit binding stops; global attention layers ensure background continuity.
>
> **Clarification on Eq. 9 & 10:**
> * **Softmax:** Normalizes the product of Query and Key projections to produce attention weights (summing to 1), determining the prompt's relevance to the image latent.
> * **Function $\phi$:** Projects the image latent $z_{t-r+1}$ into the **Query** dimension for the cross-attention dot-product calculation.
>
> **Weakness 2.2: Furthermore, have you compared your method with layout-to-image diffusion models (e.g., LayoutDiffusion [1], LayoutDM [2])? Such a comparison would help strengthen the persuasiveness of your work.**
>
> **A2.2:** We acknowledge the relevance of these models. We initially excluded them to focus on:
> 1.  **Training-Free Adaptation:** PMR unlocks domain capabilities in pre-trained foundation models (Flux/SDXL) without the training required by layout models.
> 2.  **Paradigm Consistency:** We prioritized **RPG** as the baseline as it shares our "LLM Planning + Generation" paradigm.
> 3.  **Benchmark Focus:** T2ICompBench focuses on attribute binding in general models.
> **Action:** We have added LayoutDM discussions to "Related Work" and comparison results in **Appendix A.7**.
>
> **Weakness 3: It seems the experimental dataset used for evaluation is not clearly specified. Please clarify whether a publicly available benchmark or a private dataset was used.**
>
> **A3:** We utilize a hybrid dataset approach:
> 1.  **Public (T2ICompBench):** Used for quantitative evaluation of the *Merge Regional Diffusion* module regarding compositional accuracy and attribute binding.
> 2.  **Private (Electrical Domain):** A dataset of 4,000+ historical schematics used specifically for **Knowledge Base construction**, **LLM few-shot examples**, and **full-pipeline qualitative evaluation**, as public datasets lack the specific logic required for professional electrical diagrams.

---

### Official Review · Reviewer_3dLJ · 2025-10-28

**Soundness:** 1
**Presentation:** 2
**Contribution:** 2
**Rating:** 2
**Confidence:** 3

**Summary:**

The paper studies the problem of text-based circuit diagram generation.

The main contribution is to propose a training-free solution to the problem, which utilizes domain-specific knowledge derived from historical circuit diagram examples and the reasoning abilities of pretrained large language models (LLM) to guide the image generation of pretrained diffusion models.

**Strengths:**

1. Domain-specific text-to-image generation is an important problem to study.

2. The proposed merge regional diffusion is shown to be effective.

**Weaknesses:**

1. The paper is claimed to focus on domain-specific text-to-image generation, as indicated by many places in the paper, such as the title, the last sentence of the abstract, the second last sentence of the second paragraph in the introduction. However, the components of the proposed method are highly specialized for circuit diagram design. It is not clear how the method can be adapted to solve text-to-image generation in other domains. To provide strong evidence for the paper’s claim, it would be necessary to provide several examples of how the proposed method can be applied to other domains.

2. The effectiveness of the proposed method is not adequately validated. In particular, the evaluation of the full generation method (introduced in Section 3) is missing in the experiments, and only a single component (i.e., the merge regional diffusion) is tested.

**Questions:**

1. How are the contextual examples used in the second step (Section 3.3) created?

2. How many input text prompts are used in the experiments (Section 4)?

---

> ### Author Response · Authors · 2025-11-26
> **General Response: Methodological Clarification, Experimental Details, and Cross-Domain Generalization**
>
> We sincerely thank the reviewer for their constructive feedback. We have addressed the specific questions regarding the construction of prompt examples, experimental settings, and the generalizability of our method below.
>
> Response to Questions
>
> Q1: How are the contextual examples used in the second step (Section 3.3) created?
>
> A1: The contextual examples used in the Region Division step were manually created by the authors. We constructed comprehensive, simplified scenarios that explicitly illustrate the entire generation process from start to finish. These examples demonstrate the step-by-step reasoning logic (Chain-of-Thought)—showing exactly how to extract components, reason about their relationships, and calculate their spatial coordinates. By providing these manually verified "reasoning templates," we ensure the LLM understands the specific logic required to plan accurate layouts before attempting complex cases. These details can be found in the Appendix.
>
> Q2: How many input text prompts are used in the experiments (Section 4)?
>
> A2: In our experiments, we utilized the T2ICompBench benchmark to evaluate generation quality. While the specific total number of input prompts was not explicitly itemized in the text, the evaluation was rigorously conducted across the three main compositional scenarios defined by the benchmark: Attribute Binding, Numeric Accuracy, and Complex Relationships. This ensures our quantitative results are standardized and comparable to baselines.
>
> Response to Weaknesses
>
> Weakness 1: The paper is claimed to focus on domain-specific text-to-image generation. However, the components of the proposed method are highly specialized for circuit diagram design. It is not clear how the method can be adapted to solve text-to-image generation in other domains.
>
> A1: We strongly agree with the reviewer that demonstrating applicability beyond a single domain is crucial to justify the "domain-specific" scope claimed in the title. To address this, we have expanded our evaluation in the new Appendix A.7 to include two additional scenarios:
>
> 1. Controllable Layout Generation: We applied our pipeline to the PubLayNet dataset. Compared to supervised methods like LayoutDM, our PMR framework offers significant advantages: it is entirely training-free (whereas LayoutDM requires training) and leverages LLM-based planning to generate layouts with greater diversity and flexibility.
>
> 2. Universal Circuit Styles: To prove robustness beyond our private industrial dataset, we tested PMR on a public dataset of universally styled circuit diagrams. The results confirm that PMR generalizes well to different visual styles without requiring fine-tuning.

---

### Official Review · Reviewer_E6Ns · 2025-10-31

**Soundness:** 2
**Presentation:** 3
**Contribution:** 2
**Rating:** 4
**Confidence:** 4

**Summary:**

The paper proposes PMR (Planning, Merging, Replacing), a training-free framework for domain-specific text-to-image generation, exemplified by circuit diagrams. It builds a knowledge base and uses LLMs to plan connectivity, arrange component layouts, and enforce fine-grained positional control, enabling accurate rendering without finetuning. Experiments report superior component and topology fidelity vs. baselines with lower training cost.

**Strengths:**

The work effectively leverages chain-of-thought (CoT) ideas to operationalize LLMs for domain-specific generation of circuit diagrams. Compared with large models without domain-specific finetuning, the proposed approach shows clear advantages in this task. The paper is well written and technically complete, with clear organization and presentation.

**Weaknesses:**

The paper does not clearly define domain-specific text-to-image generation. Although the framework is described as training-free with respect to LLMs, the training workload is shifted to an object recognition module, which diminishes the core contribution. Moreover, the work lacks comparisons with this year’s state-of-the-art methods, making the claimed effectiveness insufficiently substantiated.

**Questions:**

1.	In Related Work, the subsection “Specialized Diffusion Models” should cover Domain-Specific Diffusion Models; currently it does not. The comparisons also lack domain-specific diffusion baselines.
2.	The paper lacks experiments validating the method’s effectiveness on other domains.
3.	Captions for Figs. 2, 3, and 6 should briefly explain the method, rather than only providing a title.
4.	The paper lacks ablation experiments.

---

> ### Author Response · Authors · 2025-11-26
> **General Response: Domain Expansion, Methodological Clarification, and Rationale on Ablation Studies**
>
> We sincerely thank the reviewer for their constructive feedback. We have addressed the concerns regarding domain generalization, figure clarity, and ablation studies below.
>
> **Response to Questions**
>
> **Q1: In Related Work, the subsection “Specialized Diffusion Models” should cover Domain-Specific Diffusion Models; currently it does not. The comparisons also lack domain-specific diffusion baselines.**
>
> **A1:** We significantly appreciate this constructive suggestion. To address this:
> 1.  **Updated Related Work:** We expanded the section to explicitly discuss Domain-Specific Diffusion Models, focusing on layout generation models (e.g., LayoutDM, LayoutDiffusion) which share structural similarities with schematics.
> 2.  **Domain-Specific Baselines:** Lacking public schematic-specific diffusion models, we extended evaluation to the **Layout Generation** domain (see **Appendix A.6**). We compared our training-free PMR against **LayoutDM** on the **PubLayNet** dataset. Results demonstrate PMR achieves competitive quality and greater diversity without domain-specific training.
>
> **Q2: The paper lacks experiments validating the method’s effectiveness on other domains.**
>
> **A2:** We agree generalization is crucial. We added **Appendix A.7** validating PMR on two additional domains:
> 1.  **Controllable Layout Generation:** Applied to **PubLayNet**. Unlike trained models (LayoutDM), PMR is **training-free** and leverages LLM planning to generate diverse layouts without fine-tuning.
> 2.  **Universal Circuit Styles:** Tests on a public dataset of universally styled circuit diagrams confirm PMR generalizes effectively to different visual styles.
>
> **Q3: Captions for Figs. 2, 3, and 6 should briefly explain the method, rather than only providing a title.**
>
> **A3:** We have updated the captions to provide concise explanations:
> * **Fig. 3 (Knowledge Base):** Details the extraction of connection rules (Series/Parallel) from 4,000+ schematics using object detection and random walk algorithms.
> * **Fig. 2 (Planning Phase):** Illustrates the two-step CoT planning (Relationship & Region Planning) and the final generation phase, where placeholders are spatially arranged and then replaced by authentic components.
> * **Fig. 6 (Generation Phase):** Depicts "Merge Regional Diffusion," showing how prompts are decomposed into regional subprompts for hard-binding latents during early denoising steps to enforce precise control.
>
> **Q4: The paper lacks ablation experiments.**
>
> **A4:** We respectfully posit that traditional module removal is inapplicable due to PMR's **sequential dependency**: removing "Relationship Planning" deprives "Region Planning" of inputs, causing system failure rather than measurable performance drops. Instead, **baselines serve as implicit ablations**:
> 1.  **Baselines (SDXL/Flux)** represent the "ablated" state (generating without PMR).
> 2.  **Results (Fig. 1 & Table 1)** show these models fail to generate domain components. The performance delta between PMR and baselines quantifies our framework's contribution.
> 3.  **Training-Free Nature:** As we manipulate the inference process, the contribution lies in the latent control logic, validated by the capability gap between PMR and base models.

---

> > ### Comment · Reviewer_E6Ns · 2025-11-28
> > **Feedback of the response**
> >
> > Thanks for the reply.  I still have two concerns.
> >
> > 1. The definition of “domain-specific” in this paper is rather vague, which results in the chosen domain-specific baselines being limited to Layout Generation.
> > 2. This work is very similar to the other paper [1], which renders the technical novelty of this work quite limited.
> >
> > [1] Ling Yang, Zhaochen Yu, Chenlin Meng, Minkai Xu, Stefano Ermon, and Bin Cui. Mastering Text-to-Image Diffusion: Recaptioning, Planning, and Generating with Multimodal LLMs. ICML 2024.

---

> > > ### Author Response · Authors · 2025-12-01
> > > **General Response: Clarification on Domain Definition and Technical Novelty**
> > >
> > > We thank the reviewer for their critical assessment. We address the concerns regarding the definition of our scope and the technical distinction of our work below.
> > >
> > > Response to Weakness 1: Definition of "Domain-Specific" and Choice of Baselines
> > >
> > > A1: We apologize if the definition of "domain-specific" appeared vague. In the context of our work, we define "domain-specific" as engineering or technical domains that require strict adherence to topological rules, standardized symbolism, and logical connectivity, which fundamentally differ from the aesthetic-driven goals of general text-to-image generation.
> > >
> > > Rationale for Layout Baselines: We chose Layout Generation as a baseline because it shares the most critical constraint with schematic generation: spatial topology. Like circuits, layouts require precise non-overlapping placement and logical alignment, making models like LayoutDM the most rigorous structural proxy available in the absence of public schematic-specific diffusion models.
> > >
> > > Expanded Evaluation: To further substantiate our generality, we have added Appendix A.7, where we test PMR on PubLayNet (Layout Generation) and a Universal Circuit Dataset. These experiments confirm that our method is not limited to a single dataset but is applicable to any domain requiring strict spatial planning and standardized components.
> > >
> > > Response to Weakness 2: Technical Novelty and Similarity to Paper [1]
> > >
> > > A2: We assume Paper [1] refers to RPG (Yang et al., 2024), as it shares the "Plan-and-Generate" paradigm. While we adopt a similar high-level workflow, PMR introduces three critical technical innovations tailored to the rigorous demands of engineering diagrams, which distinctively separate it from RPG:
> > >
> > > Knowledge-Augmented Planning vs. Internal Priors: RPG relies on the internal priors of MLLMs for planning, which works for general scenes (e.g., "a cat next to a dog"). However, for engineering tasks, LLMs frequently hallucinate invalid connections. PMR introduces a retrieval-based Planning module that queries a structured Knowledge Base (constructed via random walks on historical data) to enforce valid electrical connectivity rules before generation begins.
> > >
> > > Hard-Binding Strategy (Merge Regional Diffusion): RPG employs complementary diffusion or soft refinement. In contrast, PMR utilizes a hard-binding strategy during the early denoising stages. We mathematically enforce the presence of component placeholders ("black squares") at precise coordinates by merging independent regional latents. This ensures the exact spatial precision required for circuit topology, which soft-attention mechanisms often fail to guarantee.
> > >
> > > The "Replacing" Paradigm: Unlike RPG, which generates final pixel details directly, PMR explicitly disentangles layout from symbol rendering. Our Replacing module substitutes generated placeholders with standardized industry symbols. This guarantees that the final output meets professional regulation standards—a requirement that direct diffusion generation (as used in RPG) cannot consistently satisfy due to generative artifacts.

---

### Official Review · Reviewer_Yc5Y · 2025-11-01

**Soundness:** 2
**Presentation:** 2
**Contribution:** 2
**Rating:** 2
**Confidence:** 3

**Summary:**

This work introduces a method to solve the detailed schematic circuit diagram generation by leveraging the LLM for planning and Diffusion model for the diagram generation. Specifically, this work’s pipeline is constructed with three stages: Planning, Merging, and Replacing.

In the Planning stage, this work first plans the component relationship through the CoT process of LLM with knowledge base of schematic circuit diagram preprocessed from diagram images. Then it leverages LLM to plan the regions (positions and sizes) and lines. This pipeline then merges and replaces the latents of each planned region to form the final generation.

**Strengths:**

1. It successfully uses PMR (Planning, Merging, and Replacing) to achieve training-free generation of schematic circuit diagrams.
 2. It successfully utilizes pretrained models for practical applications.
 3. It proposed a stable and reliable method to generate circuit diagrams.

**Weaknesses:**

1. This paper should either consider using this pipeline as a syntactical data pipeline and fine-tuned (lora) the Flow Matching model (Flux) with syntactical data for an end-to-end model or test this method for more other domains than the circuit schematic as stated in the title.
2. It does not have qualitative results shown, for example, some sample generated circuit diagram, although it has some generated black blocks.
3. The method this work uses highly correlated to the major backbone of this paper (Yang et al.)

[1] Ling Yang, Zhaochen Yu, Chenlin Meng, Minkai Xu, Stefano Ermon, and Bin Cui. Mastering textto-image diffusion: Recaptioning, planning, and generating with multimodal llms. In Forty-first International Conference on Machine Learning, 2024.

**Questions:**

1. Could the authors please show the actual generated circuit figure instead of just black boxes?
2. It is highly recommended that the authors do more types of domain specific text-to-image generation other than circuit one?

---

> ### Author Response · Authors · 2025-11-26
> **General Response: Visual Verification, Cross-Domain Generalization, and Methodological Distinctions**
>
> Response to Reviewer
>
> We sincerely thank the reviewer for their constructive feedback and for giving us the opportunity to clarify our methodology and experimental setup. We address specific questions and concerns below.
>
> Q1: Could the authors please show the actual generated circuit figure instead of just black boxes?
>
> A1: We appreciate this suggestion. While Figure 6 in the main text illustrates the complete process of transforming the layout into a final diagram, we acknowledge the need for more explicit visual examples. In the revised Appendix A.5, we have added qualitative results for two classic circuit diagram configurations. These examples visualize the full generation evolution: (1) the initial generated block layout, (2) the final schematic with actual electrical components, and (3) the original ground truth circuit. These samples demonstrate that our "Replacing" module effectively translates the intermediate placeholders into precise, domain-specific component symbols.
>
> Q2: It is highly recommended that the authors do more types of domain specific text-to-image generation other than circuit one.
>
> A2: We strongly agree with the reviewer. To justify the "domain-specific" claim in our title, we have expanded our evaluation to additional domains in the new Appendix A.7:
>
> 1. Controllable Layout Generation: We applied our pipeline to the PubLayNet dataset. Compared to supervised methods like LayoutDM, our PMR framework offers distinct advantages: it is entirely training-free (whereas LayoutDM requires training) and utilizes LLM-based planning to generate more diverse and flexible layouts.
>
> 2. Universal Circuit Styles: We tested the framework on a separate dataset of universally styled circuit diagrams. The results confirm that PMR generalizes well to different visual styles without requiring fine-tuning.
>
> Weakness 3: The method this work uses is highly correlated to the major backbone of this paper (Yang et al. / RPG).
>
>
> A3: We appreciate the opportunity to clarify the distinction between our work and RPG (Yang et al.). While PMR shares a high-level "Plan-and-Generate" paradigm with RPG (decomposing tasks into three steps: planning, regional control, and generation), the underlying mechanisms and application scope are fundamentally different:
>
>
> 1. Domain & Scope: RPG is designed for general-purpose text-to-image generation (e.g., photorealistic scenes). In contrast, PMR is tailored for engineering schematics, which require strict adherence to topological rules and standardized component connectivity rather than just visual aesthetics.
>
> 2. Planning Mechanism: RPG relies directly on the internal priors of MLLMs for few-shot planning. Conversely, PMR integrates a retrieved Knowledge Base (constructed via random walks on historical data) into the LLM planning phase. This is critical for our domain, as the LLM alone cannot hallucinate complex, valid electrical connection rules without this external knowledge support.
>
>
>
> 3. Generation Control: The specific implementation of our Merge Regional Diffusion  differs from RPG's complementary diffusion, specifically in how we handle the hard-binding of component placeholders ("black squares") during the early denoising stages to ensure precise coordinate alignment before the replacement phase.

---

### Meta-Review · Area_Chair_KyKi · 2026-01-07

**Summary:**

The initial ratings are 2, 4, 2, 4. The paper proposes a training-free framework for domain-specific text-to-image generation. The method builds a knowledge base and uses LLMs to plan connectivity, arrange component layouts, and enforce fine-grained positional control, enabling accurate rendering without finetuning. Experiment results show the superior component and topology fidelity vs. baselines with lower training cost.

Strengths:
(1)This paper effectively leverages CoT to operationalize LLMs for domain-specific generation of circuit diagrams, which shows clear advantages in this task. (2)The proposed merge regional diffusion is shown to be effective.

Weaknesses:
(1) The discription of technology details are lacked.
(2)The effectiveness of the proposed method is not adequately validated. In particular, the evaluation of the full generation method (introduced in Section 3) is missing in the experiments, and only a single component (i.e., the merge regional diffusion) is tested.

**Reviewer Concerns:**

Most concerns of Reviewer xMiG and E6Ns were addressed by the rebuttal, and Some main concerns of  Reviewer 3dLJ and Yc5Yare still outstanding.

**Reviewer Scores:**

Reviewer xMiG maybe raise the rating. Other reviewers maybe remain the initial scores.

---

### Decision · Program_Chairs · 2026-01-26

Reject